# Change in therapeutic management after the EndoPredict assay in a prospective decision impact study of Mexican premenopausal breast cancer patients

Cynthia Villarreal-Garza[1,2]*, Edna Anakarenn Lopez-Martinez[1], Zuratzi Deneken-Hernandez[3], Antonio Maffuz-Aziz[3], Jose Felipe Muñoz-Lozano[1], Regina Barragan-Carrillo[1], Pier Ramos-Elias[1], Brizio Moreno[1], Hector Diaz-Perez[1], Omar Peña-Curiel[1], Jose de Jesus Curiel-Valdez[4], Veronica Bautista-Piña[3]

1 Breast Cancer Center, Hospital Zambrano Hellion, Tecnologico de Monterrey, Nuevo Leon, Mexico,
2 Research and Breast Cancer Department, Instituto Nacional de Cancerologia, Mexico City, Mexico,
3 Fundacion de Cancer de Mama (FUCAM), Mexico City, Mexico, 4 Grupo Diagnostico Laboratorio Clinico y Patologia, Mexico City, Mexico

* cynthiavg@gmail.com

## Abstract

### Objective

To evaluate the change in adjuvant therapeutic decision in a cohort of young women with breast cancer discussed by a multidisciplinary team, before and after EndoPredict testing.

### Patients and methods

99 premenopausal women with hormone receptor-positive, HER2-negative, T1-T2, and N0-N1 breast cancer were included. Clinicopathological characteristics were recorded and cases were presented in a multidisciplinary tumor board. Consensual therapeutic decisions before and after EndoPredict results were registered. Medical records were reviewed at six-month follow-up to determine physicians' adherence to therapeutic recommendations. Pearson chi-square and McNemar's tests were used to analyze differences between groups and changes in treatment recommendations, respectively.

### Results

Median age at diagnosis was 43 years. The most frequent tumor size was pT2 (53.5%) and 27% of patients had 1–3 positive lymph nodes. 46% of patients had a low-risk EPclin result. Nodal status and tumor grade were significantly associated with EPclin result (p < .00001 and p = .0110, respectively), while Ki67 levels and age ≤40 years were not. A change in chemotherapy decision was registered in 19.2% of patients (p = .066), with the greatest impact in de-escalation (9% net reduction). A change in chemotherapy or endocrine therapy regimen was suggested in 19% and 20% of cases, respectively, after EPclin results were available. A significant difference was found in the median EPclin score between patients with a low- vs. high-intensity chemotherapy and endocrine therapy regimen

**Data Availability Statement:** All relevant data are within the paper and its Supporting Information files.

**Funding:** The author(s) received no specific funding for this work. EndoPredict tests were donated by Myriad Genetics. Myriad Genetics did not play any role in the study design, data collection and analysis, decision to publish, or preparation of the manuscript.

**Competing interests:** The authors have declared that no competing interests exist.

recommendation (p = 0.049 and p = 0.0001, respectively). Tumor board treatment recommendation adherence with the EndoPredict result was 95% and final treatment adherence to EPclin result was 93%.

## Conclusions

The EndoPredict test successfully assisted the clinical decision-making process in premenopausal patients, with a clinically significant change in overall decision-making, with the greatest impact seen in chemotherapy reduction, and a high rate of therapeutic adherence.

## Introduction

Choosing the appropriate treatment for breast cancer (BC) patients with hormone-receptor (HR)-positive, HER2-negative, early disease can be a thorough and challenging process that requires the proper balancing of possible therapy benefits against the risk of potential side effects.[1,2] In young women, decision-making is even more difficult as it is deeply influenced by age, resulting in the prescription of aggressive systemic therapy even in tumors with low-risk clinical features.[3–5] These intensive and prolonged systemic treatments often lead to considerable morbidity and significant psychosocial repercussions.[6–10]

Therefore, the medical community should strive to identify young patients who will not benefit from chemotherapy (CT) and could be treated with endocrine therapy (ET) alone. Gene expression profiling tests have been developed to provide clinicians with additional tools to aid them in the decision-making process, especially in situations where the benefit of adjuvant CT is equivocal.[1,11,12] These tests estimate the risk of distant recurrence in patients with HR-positive, HER2-negative BC with 0–3 positive lymph nodes. However, to date, clinical trials validating the use of genomic signatures in the clinical setting have included a limited number of young patients, thus restricting the extrapolation of results to this population.[13] Likewise, although some studies have evaluated the impact of these assays in oncological decision-making,[14–18] most of the assessed patients have been postmenopausal women and no study has focused exclusively in young women with BC.

EndoPredict, a multigene expression profiling test that predicts the likelihood of distant recurrence in patients with HR-positive, HER2-negative BC treated with adjuvant ET,[16] has been proven to be highly prognostic in node-negative or node-positive disease for early and late recurrence,[19–21] and also predicts benefit from adjuvant CT.[22] Additionally, it has been shown to be an independent prognostic parameter both in premenopausal and postmenopausal women treated with CT.[23]

The EPclin score, the final result of the EndoPredict test, takes into account molecular parameters and relevant clinical characteristics such as tumor size and nodal status, and can aid clinicians in the therapeutic decision-making process by classifying a patient as having a low- or high-risk of distant recurrence.[16,17,24] The aim of this study was to evaluate the change in therapeutic decision regarding adjuvant treatment in a cohort of young women with BC discussed by a multidisciplinary team, before and after the EndoPredict assay was performed.

## Study design

Premenopausal patients with HR-positive, HER2-negative, T1-T2, and N0-N1 BC, who had not undergone systemic treatment, were eligible to participate in this study. Patients were classified as premenopausal if they had regular menses or serum FSH and estradiol levels in premenopausal ranges in case of amenorrhea <12 months of duration or irregular menses during

the last year. Women who met selection criteria were prospectively accrued between June 2016 and August 2018 at three specialized BC centers in Mexico: "Hospital San José" and "Hospital Zambrano Hellion" in Nuevo Leon, and "Fundación de Cáncer de Mama (FUCAM)" in Mexico City. EndoPredict tests were performed locally and results were available in a median of 18 calendar days. The study was approved by the institutional ethics and research committees of "Escuela de Medicina del Instituto Tecnológico y de Estudios Superiores de Monterrey": "Comité de Ética en Investigación" and "Comité de Investigación". Written consent was obtained from all patients.

Clinicopathological characteristics were recorded and each case was presented at each institution in a multidisciplinary tumor board comprised by medical oncologists, breast surgeons, gynecological oncologists, and radiation oncologists. Initial consensual therapeutic decisions were agreed on prior to EndoPredict results, and a second consensus was reached once results were available. The therapeutic recommendation was presented to the patient after this final decision.

The type of adjuvant treatment was decided according to an estimation of the risk of recurrence based on the clinical, pathological and molecular information of each particular case. More intensive or prolonged schemes were considered in high-risk patients, while less intensive or shorter ones were proposed in low-risk patients. Patients' medical records were reviewed at six-month follow-up to determine physicians' adherence to the tumor board recommendation. Patients' total follow-up and time to recurrence were calculated according to the time between accrual and last visit and the time between accrual and recurrence documentation, respectively.

The primary objective was to determine the change in decision-making by the tumor board regarding the use of adjuvant CT before and after disclosure of the EndoPredict result. Secondary objectives were to describe CT and ET recommended regimen changes before and after EndoPredict and to evaluate physicians' adherence with tumor board therapeutic consensus regarding CT use. Pearson chi-square was used to analyze low- or high-risk EPclin results between groups according to clinicopathological features such as age, nodal status, grade, and Ki67 levels. McNemar's test was used to evaluate the change in treatment recommendations before and after EndoPredict testing. The Mann-Whitney U test was used to compare the median EPclin score between low- and high-intensity CT and ET regimens.

For CT regimens, docetaxel + cyclophosphamide (TC) was considered a low-intensity treatment, while sequential anthracyclines + cyclophosphamide followed by taxanes regimen (AC-T) was classified as a high-intensity treatment. Regarding ET, tamoxifen for 5 years (y) and tamoxifen for 2-3y followed by an aromatase inhibitor (AI) until completing 5y were considered low-intensity regimens, while tamoxifen for 10y, ovarian function suppression (OFS) with a gonadotropin-releasing hormone (GnRH) analogue and either tamoxifen or an AI for 5y, were classified as high-intensity treatments.

## Results

### Cohort description

A total of 99 consecutive premenopausal women were included in this study. Patient demographics and clinical characteristics are detailed in Table 1. Median age at diagnosis was 43y, with 32.2% of patients aged ≤40y. The most frequent tumor size was pT2 (53.5%). Furthermore, 27.3% of patients had 1–3 positive lymph nodes including micrometastasis. Stages at diagnosis were IA, 38.4%; IB, 2%; IIA, 37.4%; and IIB, 22.2%. The most common tumor subtype was invasive ductal carcinoma (89%), followed by invasive lobular carcinoma (4%) and mucinous carcinoma (3%). Pathologic characteristics showed intermediate-grade tumors in 74% of patients and low Ki67 (<20%) in half of the cohort (Tables 1 and 2).

**Table 1. Patients' clinical and pathological characteristics.**

| Total number of cases | | 99 (100%) |
|---|---|---|
| **Patient age (years)** Median: 43 | ≤35 | 10 (10.1%) |
| | 36–40 | 22 (22.2%) |
| | 41–45 | 36 (36.4%) |
| | 46–50 | 28 (28.3%) |
| | >50 | 3 (3%) |
| **Size (T)** | pT1a | 1 (1%) |
| | pT1b | 10 (10.1%) |
| | pT1c | 35 (35.4%) |
| | pT2 | 53 (53.5%) |
| **Lymph Nodes (N)** | pN0 | 71 (71.7%) |
| | pN1mic | 2 (1%) |
| | pN1 | 26 (26.3%) |
| **Clinical Stage** | IA | 38 (38.4%) |
| | IB | 2 (2%) |
| | IIA | 37 (37.4%) |
| | IIB | 22 (22.2%) |
| **Tumor subtype** | Invasive ductal carcinoma | 88 (89%) |
| | Invasive lobular carcinoma | 4 (4%) |
| | Mucinous carcinoma | 3 (3%) |
| | Other | 3 (3%) |
| | Not available | 1 (1%) |
| **Tumor grade** | I | 11 (11.1%) |
| | II | 73 (73.8%) |
| | III | 11 (11.1%) |
| | Not available | 4 (4%) |
| **Ki67** | <20% | 44 (44.4%) |
| | ≥20% | 44 (44.4%) |
| | Not available | 11 (11.1%) |
| **Estrogen receptor** | Positive | 99 (100%) |
| | Negative | 0 (0%) |
| **Progesterone receptor** | Positive | 89 (90%) |
| | Negative | 6 (6%) |
| | Not available | 4 (4%) |
| **EPclin score** | Low-risk | 46 (46.5%) |
| | High-risk | 53 (53.5%) |

## Association of clinical and pathological factors with EndoPredict test result

A total of 46 patients (46.5%) had a low-risk EPclin result. Notably, 38% of patients in the node-negative group had a high-risk result, while 8% with N1 status had a low-risk score (p < .00001). When comparing results by age group, 34% of ≤40y-old patients had a low-risk EPclin, compared with 52% in older patients (p = .09). Tumor grade was significantly associated with the EPclin result, with 82% of low-grade tumors being categorized as low-risk, while 82% of high-grade tumors had a high-risk result (p = .0110). Intermediate-grade tumors were not predictive of EPclin results with a nearly 50–50 split between low- and high-risk categories. No significant association was found between EPclin and Ki67 levels, as 41% of low Ki67 tumors were classified as high-risk, while 36% of high Ki67 tumors had a low-risk score (p = .0548).

**Table 2. Patients' clinical and pathological characteristics by EPclin risk classification.**

| Total number of cases | | Low risk 46 (100%) | High risk 53 (100%) |
|---|---|---|---|
| **Patient age (years)**Median: 43 | ≤35 | 5 (10.9%) | 5 (9.4%) |
| | 36–40 | 6 (13%) | 16 (30.2%) |
| | 41–45 | 19 (41.3%) | 17 (32.1%) |
| | 46–50 | 14 (30.4%) | 14 (26.4%) |
| | >50 | 2 (4.3%) | 1 (1.9%) |
| **Size (T)** | pT1a | 1 (2.2%) | 0 (0%) |
| | pT1b | 9 (19.6%) | 1 (1.9%) |
| | pT1c | 22 (47.8%) | 13 (24.5%) |
| | pT2 | 14 (30.4%) | 39 (73.6%) |
| **Lymph Nodes (N)** | pN0 | 44 (95.7%) | 27 (50.9%) |
| | pN1mic | 0 (0%) | 2 (3.8%) |
| | pN1 | 2 (4.3%) | 24 (45.3%) |
| **Clinical Stage** | IA | 30 (65.2%) | 8 (15.1%) |
| | IB | 1 (2.2%) | 1 (1.9%) |
| | IIA | 13 (28.3) | 24 (45.3%) |
| | IIB | 2 (4.3%) | 20 (37.7%) |
| **Tumor subtype** | Invasive ductal carcinoma | 41 (89.1%) | 47 (88.7%) |
| | Invasive lobular carcinoma | 1 (2.2%) | 3 (5.7%) |
| | Mucinous carcinoma | 3 (6.5%) | 0 (0%) |
| | Other | 0 (0%) | 3 (5.7%) |
| | Not available | 1 (2.2%) | 0 (0%) |
| **Tumor grade** | I | 9 (19.6%) | 2 (3.8%) |
| | II | 34 (73.9%) | 39 (73.6%) |
| | III | 2 (4.3%) | 9 (17%) |
| | Not available | 1 (2.2%) | 3 (5.7%) |
| **Ki67** | <20% | 26 (56.5%) | 18 (34%) |
| | ≥20% | 16 (34.8%) | 28 (52.8%) |
| | Not available | 4 (8.7%) | 7 (13.2%) |
| **Estrogen receptor** | Positive | 46 (100%) | 53 (100%) |
| | Negative | 0 (0%) | 0 (0%) |
| **Progesterone receptor** | Positive | 43 (93.5%) | 46 (86.8%) |
| | Negative | 1 (2.2%) | 5 (9.4%) |
| | Not available | 2 (4.3%) | 2 (3.8%) |

## Impact of the EndoPredict test result on adjuvant treatment decision

A change in CT decision was registered in 19/99 patients (19.2%; p = .066, McNemar test), with the greatest impact in CT de-escalation (Table 3 and Fig 1). Before having the test result, CT was recommended to 68% of patients; while post-test, CT was recommended in 59% of cases, resulting in a net reduction of CT recommendation of 9%. Net reduction was 13% in the N0 group (55% pre-test vs. 42% post-test; p = .066) and 0% in the N1 group (100% pre-test vs. 100% post-test; p = 0).

Overall tumor board treatment recommendation adherence to the EndoPredict result was 95% (recommending CT to patients with high-risk EPclin and abstaining to do so in patients with low-risk EPclin). The total population with a high-risk EPclin score was recommended to undergo CT, while 89% with a low-risk EPclin result was not. A total of five patients were recommended to undergo CT by the tumor board despite a low-risk EPclin result (Table 4).

**Table 3. Pre- and post-EndoPredict chemotherapy consensus.**

| Pre-test | | Post-test | | |
|---|---|---|---|---|
| | | No chemotherapy | Yes chemotherapy | Total |
| | No chemotherapy | 27 | 5 | 32 |
| | Yes chemotherapy | 14 | 53 | 67 |
| | Total | 41 | 58 (9% reduction) | 99 |

p = .066 for change in recommendation for chemotherapy

Furthermore, 10/53 (19%) patients who were recommended to undergo CT both pre- and post-test had a change in the suggested CT regimen. The most frequent change was from TC to AC-T, in 6/10 cases (60%) (Table 5).

Additionally, 20/99 (20.2%) patients had a change in the recommended ET regimen after the test result disclosure, with tamoxifen for 5y to tamoxifen for 10y being the most common treatment modification, seen in 9/20 cases (45%) (Table 6).

A significant difference was found in the median EPclin score between patients with a low- vs. high-intensity CT regimen recommendation (3.7 [1.3 SD] vs 4 [1.2 SD], p = 0.049) (Fig 2).

Similarly, there was also a significant difference in the median EPclin score between patients with a low- vs. high-intensity ET regimen recommendation (3 [1.0 SD] vs 3.9 [1.4 SD], p = 0.0001) (Fig 3).

Regarding attending physicians' adherence to the tumor board post-test recommendation, the suggested treatment regimen was followed in 98% of cases. In two patients, CT was not

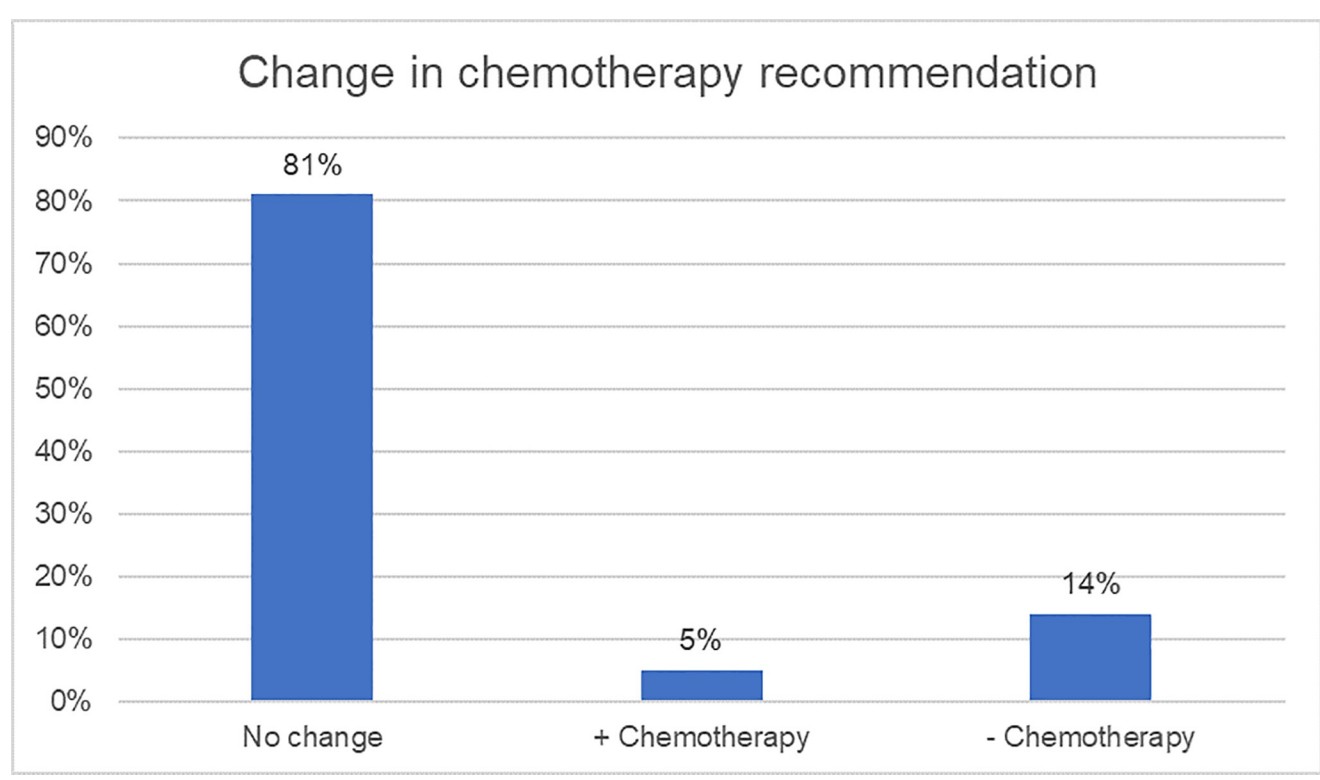

**Fig 1. Change in chemotherapy recommendation.**

**Table 4. Low-risk EPclin patients recommended to undergo chemotherapy by the tumor board.**

| Age (y) | Tumor size (mm) | pT | pN | Stage | Grade | Ki67 | EPclin | Reason for chemotherapy recommendation | Recommended chemotherapy regimen |
|---|---|---|---|---|---|---|---|---|---|
| 41 | 32 | pT2 | pN1 | IIB | Low | 5 | 2.8 | Two positive nodes | TC |
| 50 | 25 | pT2 | pN0 | IIA | Intermediate | 10 | 3.32 | Tumor size and borderline EPclin* | TC |
| 49 | 50 | pT2 | pN0 | IIB | Intermediate | 20 | 3 | Tumor size and borderline Ki67 level | TC |
| 41 | 20 | pT1c | pN0 | IA | Low | NA | 3.29 | Borderline EPclin* | AC-T |
| 39 | 15 | pT1c | pN1 | IIA | Intermediate | 5 | 3.2 | Three positive nodes, two with extracapsular extension | AC-T |

**TC:** Docetaxel + Cyclophosphamide

**AC-T:** Anthracyclines + Cyclophosphamide followed by Taxanes

**NA:** Not available

*Borderline EPclin refers to a result that was close to the 3.3 cutoff point for EPclin score–based risk stratification

recommended by the tumor board due to the patients' low-risk clinical features and a low-risk EPclin result; however, treatment was ultimately prescribed by their attending physician, who did not attend the BC multidisciplinary tumor board. In both cases, the reason for recommending CT was not documented in the patients' medical files.

Final treatment adherence to EPclin result was 93% (administering CT in high-risk EPclin and no CT in low-risk EPclin).

A comparison between CT recommendation pre- and post-EPclin result, as well as the actual treatment prescribed to patients by their treating physicians is shown in Fig 4.

## Follow-up

Patients' median follow-up time was 22.9 months. During this period, four patients presented disease recurrence, with follow-up times to recurrence of 11.7, 14.3, 36.9, and 42.4 months. Two of these patients were classified as low-risk by the EndoPredict test and had local recurrences. Of the other two patients, classified as high-risk, one had a local recurrence and the other one had a distant recurrence. All patients were alive up to their last follow-up visit.

## Discussion

This is the first study that specifically evaluates the EndoPredict assay effect in decision-making in a premenopausal BC patient cohort. Overall, we reported a higher proportion of high-risk patients in this study (54%) when compared to the previous validation trials ABCSG 6&8

**Table 5. Pre- and post-EndoPredict chemotherapy regimens.**

| Pre-EndoPredict regimen | Post-EndoPredict regimen | Number of patients (n = 53) |
|---|---|---|
| Anthracyclines + Cyclophosphamide followed by Taxanes** | AC-T** | 23 (43%) |
|  | TC* | 3 (6%) |
| Docetaxel + Cyclophosphamide* | TC* | 20 (38%) |
|  | AC-T** | 6 (11%) |
| Other | AC-T** | 1 (2%) |

**TC:** Docetaxel + Cyclophosphamide

**AC-T:** Anthracyclines + Cyclophosphamide followed by Taxanes

*Low-intensity regimen

**High-intensity regimen

**Table 6. Pre- and post-EndoPredict endocrine therapy regimens.**

| Pre-EndoPredict regimen | Post-EndoPredict regimen | Number of patients (n = 99) |
|---|---|---|
| Tamoxifen 5y* | Tamoxifen 5y* | 47 (48%) |
| | Tamoxifen 10y** | 9 (9%) |
| | Tamoxifen 2–3 y, then AI until 5 y* | 1 (1%) |
| | Tamoxifen + GnRH analogue 5 y** | 2 (2%) |
| | AI + GnRH analogue 5y** | 3 (3%) |
| Tamoxifen 10y** | Tamoxifen 10y** | 12 (12%) |
| | Tamoxifen 5y* | 2 (2%) |
| | Tamoxifen 2–3 y, then AI until 5 y* | 1 (1%) |
| Tamoxifen 2–3 y, then AI until 5 y* | Tamoxifen 2–3 y, then AI until 5 y* | 9 (9%) |
| | Tamoxifen 5y* | 1 (1%) |
| AI + GnRH analogue 5y** | AI + GnRH analogue 5y** | 9 (9%) |
| | Tamoxifen 5y* | 1 (1%) |
| Tamoxifen + GnRH analogue 5y** | Tamoxifen + GnRH analogue 5y** | 2 (2%) |

**y:** years

**AI:** aromatase inhibitor

*Low-intensity regimen

**High-intensity regimen

and TransATAC, where 37% and 41.2% of patients had a high-risk result, respectively.[24,25] These findings might reflect the higher baseline risk of our younger patients, compared to the previously mentioned trials, which only included postmenopausal patients.

Furthermore, we observed a clinically significant impact on treatment de-escalation with a 9% absolute reduction in CT recommendation. This impact is smaller than that reported in three other decision impact trials, where absolute reduction in CT recommendation ranged from 13.1% to 33%.[15–17] The reported lower overall change in treatment decision could be

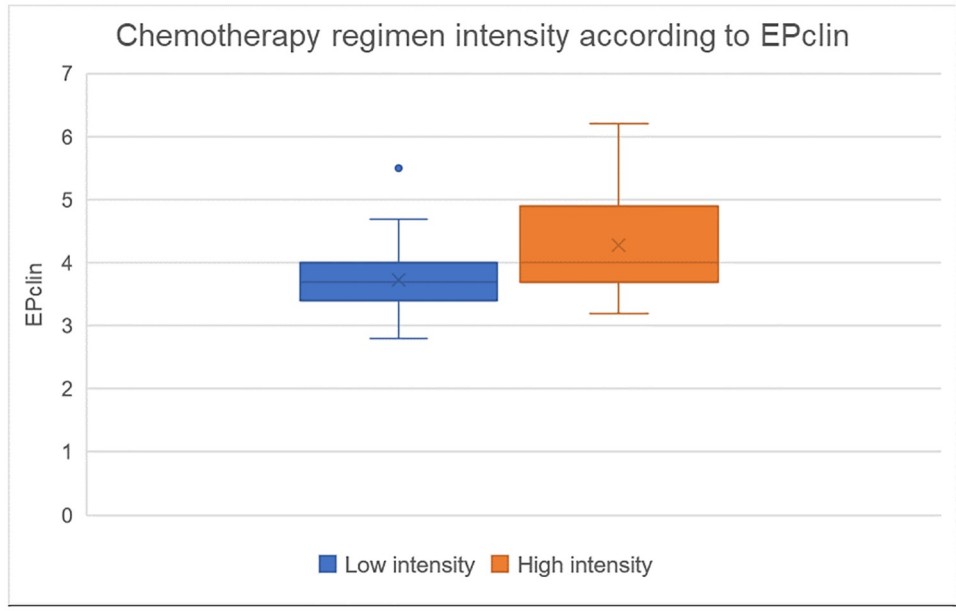

**Fig 2. Chemotherapy regimen intensity according to EPclin.**

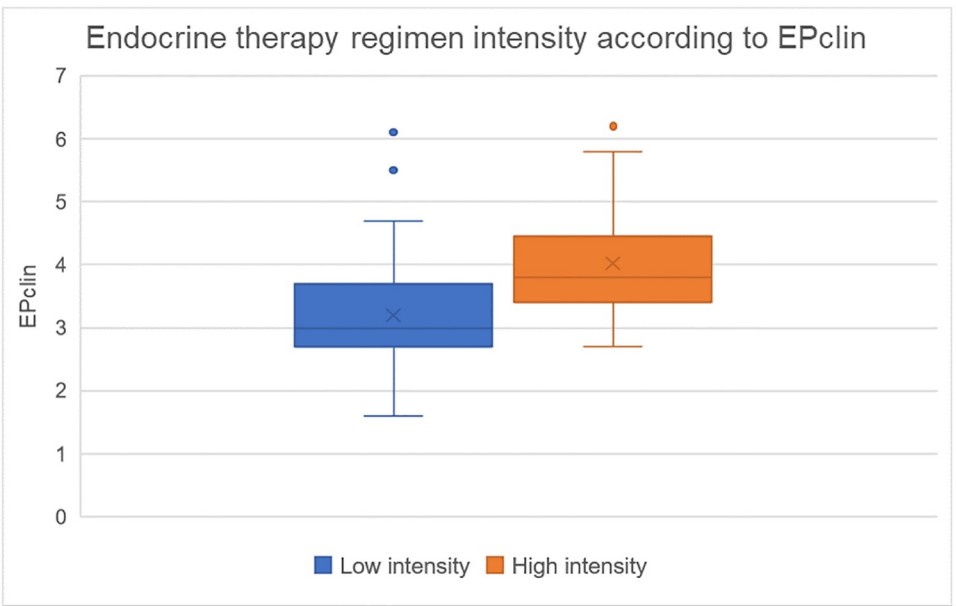

**Fig 3. Endocrine therapy regimen intensity according to EPclin.**

explained by a reliable clinical judgment by tumor boards held at academic centers for identifying high-risk pre-menopausal patients based on clinicopathological variables alone. Nevertheless, our physicians may have overtreated patients if therapeutic decisions relied on clinical characteristics only, as we found a substantial discordance between nodal status, tumor grade and Ki67 levels compared to EPclin results, which justifies the use of gene expression panels.

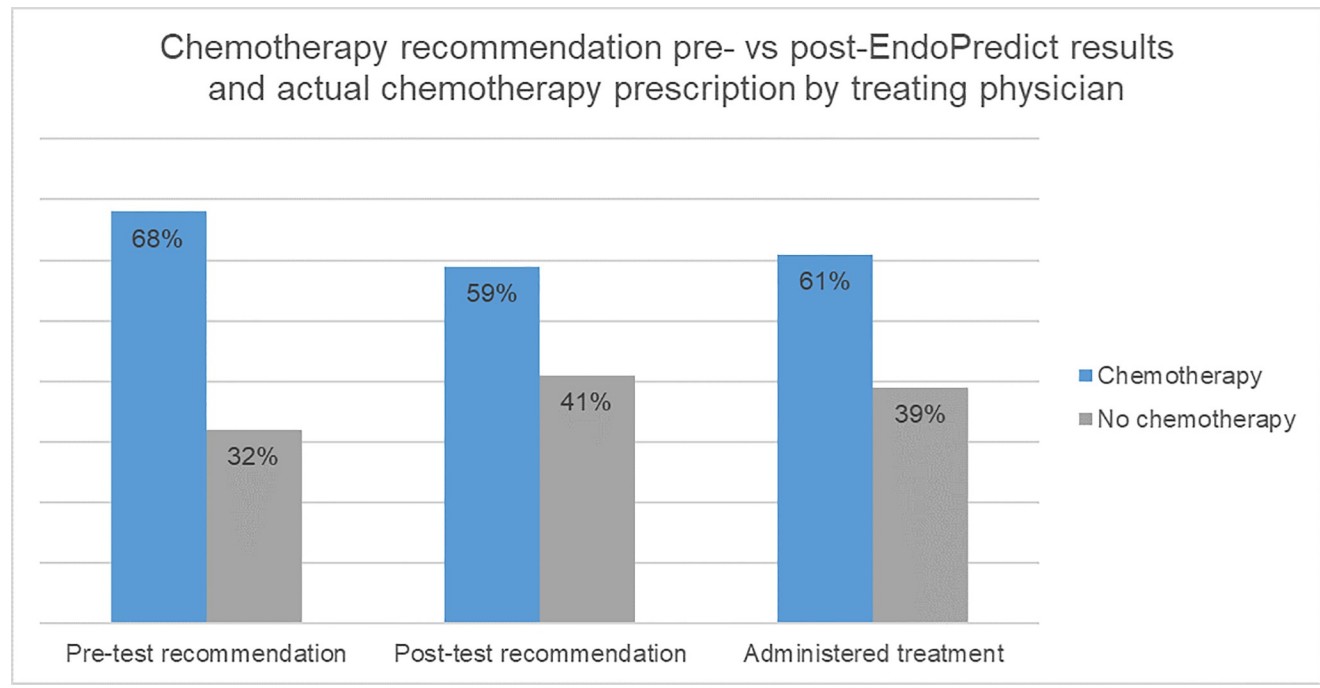

**Fig 4. Chemotherapy recommendation pre- vs. post-EndoPredict results and actual chemotherapy prescription by treating physician.**

Interestingly, an impact on the CT and ET regimen was also observed when comparing the pre- and post-test tumor board consensus. While there is no data supporting the change in the CT and ET regimen based on gene expression assay results, there is a rationale for administering less intense regimens in patients with a low-risk EPclin, especially when considering the high prognostic performance of the EndoPredict assay for early and late recurrence in both node-negative and node-positive disease.[19–21,25]

Recently, 15-y long-term outcome data showed that patients with a low-risk EPclin score treated with 5y of ET alone had a very low 5-15-y risk of distant recurrence, with a distant recurrence-free rate of 95.7%. Therefore, extension of ET up to 10y might not be necessary in this subgroup.[20] This supports the less intense ET in our patients.

Remarkably, tumor board treatment recommendation according to the EndoPredict test result (recommending CT to patients with high-risk EPclin and abstaining to do so in patients with low-risk EPclin) and final treatment adherence to the test score reached 95% and 93%, respectively. Our treatment adherence was even higher than the 85% reported previously by Fallowfield et al.[14] Additionally, to our knowledge, this is the first study that describes attending physicians' adherence to the tumor board post-test recommendation.

One of the strengths of this study is the prospective evaluation of a well-defined population of patients and recruitment from three different BC centers in Mexico. Furthermore, it is focused exclusively in premenopausal BC patients and describes how the EPclin score impacts the type of recommended CT or ET regimen as well as adding to the overall information of the performance of the EndoPredict assay.

## Conclusion

Overall, the EndoPredict assay's greatest impact was aiding in the discrimination of premenopausal patients that would not benefit from CT. While the overall CT reduction in our study was lower than that in previously published trials, it is still clinically relevant as it allowed several young patients, typically regarded as high-risk, to avoid unnecessary adverse effects that could significantly affect their quality of life.

The EndoPredict test successfully assisted the clinical decision-making process in premenopausal patients, with a high rate of therapeutic adherence and a clinically significant change in overall decision-making.

## Supporting information

**S1 Appendix. Research protocol complete database.**
(XLSX)

## Acknowledgments

We thank Dr. Servando Cardona and Dr. Mauricio Canavati, as well as the tumor board members of the Breast Cancer Center at Hospital Zambrano Hellion and Fundacion de Cancer de Mama (FUCAM), for their support and participation in this project.

## Author Contributions

**Conceptualization:** Cynthia Villarreal-Garza, Pier Ramos-Elias.

**Data curation:** Edna Anakarenn Lopez-Martinez, Zuratzi Deneken-Hernandez, Antonio Maffuz-Aziz, Jose Felipe Muñoz-Lozano, Regina Barragan-Carrillo.

**Formal analysis:** Cynthia Villarreal-Garza, Edna Anakarenn Lopez-Martinez, Jose Felipe Muñoz-Lozano.

**Funding acquisition:** Cynthia Villarreal-Garza, Pier Ramos-Elias.

**Investigation:** Cynthia Villarreal-Garza, Edna Anakarenn Lopez-Martinez, Zuratzi Deneken-Hernandez, Antonio Maffuz-Aziz, Jose Felipe Muñoz-Lozano, Regina Barragan-Carrillo, Brizio Moreno, Hector Diaz-Perez, Omar Peña-Curiel.

**Methodology:** Cynthia Villarreal-Garza, Regina Barragan-Carrillo, Pier Ramos-Elias.

**Project administration:** Cynthia Villarreal-Garza, Edna Anakarenn Lopez-Martinez, Regina Barragan-Carrillo.

**Resources:** Jose de Jesus Curiel-Valdez, Veronica Bautista-Piña.

**Supervision:** Cynthia Villarreal-Garza.

**Visualization:** Cynthia Villarreal-Garza, Edna Anakarenn Lopez-Martinez.

**Writing – original draft:** Cynthia Villarreal-Garza, Edna Anakarenn Lopez-Martinez.

**Writing – review & editing:** Cynthia Villarreal-Garza, Edna Anakarenn Lopez-Martinez, Zuratzi Deneken-Hernandez, Antonio Maffuz-Aziz, Jose Felipe Muñoz-Lozano, Regina Barragan-Carrillo, Pier Ramos-Elias, Brizio Moreno, Hector Diaz-Perez, Omar Peña-Curiel, Jose de Jesus Curiel-Valdez, Veronica Bautista-Piña.

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
