## [Decision Letter · Decision Letter 0]

7 Nov 2019

PONE-D-19-24535

Change in therapeutic management after the EndoPredict assay in a prospective decision impact study of Mexican premenopausal breast cancer patients

PLOS ONE

Dear Dr. Villareal-Garza,

Thank you for submitting your manuscript to PLOS ONE. After careful consideration, we feel that it has merit but does not fully meet PLOS ONE’s publication criteria as it currently stands. Therefore, we invite you to submit a revised version of the manuscript that addresses the points raised during the review process.

ACADEMIC EDITOR: Please insert comments here and delete this placeholder text when finished. Be sure to:

Indicate which changes are required versus recommended for acceptanceAddress any conflicts between the reviewsProvide specific feedback from your evaluation of the manuscript

We would appreciate receiving your revised manuscript by Dec 22 2019 11:59PM. To enhance the reproducibility of your results, we recommend that if applicable you deposit your laboratory protocols in protocols.io, where a protocol can be assigned its own identifier (DOI) such that it can be cited independently in the future. For instructions see: http://journals.plos.org/plosone/s/submission-guidelines#loc-laboratory-protocols

We look forward to receiving your revised manuscript.

Kind regards,

Academic Editor

PLOS ONE

Journal Requirements:

1. In your Methods section and the online statement form, please include the full name of the IRB/ethics committees that reviewed and approved this study for each hospital, including the name of the affiliated institutions. We additionally ask that you include your IRB/ethics committee approval number in your ethics statement.

Reviewers' comments:

Reviewer's Responses to Questions

**Comments to the Author**

1. Is the manuscript technically sound, and do the data support the conclusions?

Reviewer #1: Yes

Reviewer #2: Yes

Reviewer #3: Partly

2. Has the statistical analysis been performed appropriately and rigorously? 

Reviewer #1: Yes

Reviewer #2: Yes

Reviewer #3: Yes

3. Have the authors made all data underlying the findings in their manuscript fully available?

Reviewer #1: Yes

Reviewer #2: Yes

Reviewer #3: No

4. Is the manuscript presented in an intelligible fashion and written in standard English?

Reviewer #1: Yes

Reviewer #2: Yes

Reviewer #3: Yes

5. Review Comments to the Author

Reviewer #1: In this manuscript, the authors come up with EndoPredict assay in a prospective decision impact study of Mexican premenopausal breast cancer patients to change therapeutic management. The validity of EndoPredict test to assist the clinical decision-making process in premenopausal patients deserve deeper discussion. It would be helpful to observe the study longer for patient outcome and expand the sample size. Besides, the analysis of post- EndoPredict results should be illustrated more detailedly.

Reviewer #2: Villareal-Garza and co-authors present an interesting decision impact study of EndoPredict in specifically younger women. The manuscript is well written and presents interesting findings on pre- versus post-treatment deicision making in premenopausal women. As the authors state most validation and decision-impact studies have been performed in older, postmenopausal women and therefore their results is relevant for the clinical decision making in younger women.

A few minor points might improve the manuscript:

1. It would be helpful to add EPclin scores (means) or percentage low/high risk to Table 1 so the reader has all the information in one place.

2. Table 2 would benefit from adding percentage change. I am aware that this is stated in the text but with added percentages the reader has the full picture just looking at the table.

3. Perhaps a Figure would make Table 2 even more appealing to readers showing the changes in treatment between pre- and post EndoPredict testing.

4. Table 4 and 5 again please add percentages as number scan be deceiving.

5. For Table 5, I suggest putting the low intensity ET regimes, tamoxifen 5y and tamoxifen 2-3y then AI, underneath each other or to make it very clear in the table which are the low versus high intensity regimens.

6. Figures 1 and 2: I suggest showing box plots for the EPclin scores with 95% CI so it is clear to the reader what the means are and it is visually easier to see that there is a difference.

Reviewer #3: This paper evaluates the change in adjuvant therapeutic decision in a series of 99 premenopausal women with hormone receptor-positive/HER2-negative breast cancer before and after EndoPredict testing. A change in chemotherapy decision was recorded in around 19% of patients, with the greatest impact in de-escalation, with a net reduction of 9% in chemothepary recommendation. After EPclin result, a change in chemotherapy or endocrine therapy was recommended in 19% and 20% of cases, respectively. Aditionally, adherence to tumor board recommendation with the EndoPredict result was also analyzed. Overall, this is an interesting study that specifically evaluates EndoPredict result effect in decision-making in a cohort or premenopausal women with early breast cancer.

Comments.

Study design. Elegible participants (line 73). Specific selection criteria should be detailed in the text.

Line 83. The authors indicate that consensual therapeutic decisions before and after EndoPredict results were registered. However, the criteria followed when deciding the type of adjuvant treatment in each case (chemotherapy or endocrine therapy) are not specified. Please, indicate this criteria in the text.

I assume that EndoPredict test was performed centrally and not in each of the hospitals participating in the study. If so, please include this information in the text. Information regarding the time between the test request and the result of the test should also be detailed. Did the multidisciplinary tumor board make a provisional treatment decision-either to have or to omit adjuvant chemotherapy- in these cases? Did patients participate in decision making?

Cohort description (line 102). Tumor subtype (histology) should be detailed in the text/table 1. Please, also include the distribution of cases regarding estrogen receptor positivity (should be 100%!) and, more interesting as is usual more heterogeneous than estrogen receptor, progesterone receptor positivity/negativity.

The inclusion of a patient with a pT1a tumor is striking. Could you please give explain why this patient was included in the series (high tumor grade, lymph node mets…)?

Was the molecular phenotype of the tumors (mainly, luminal A and luminal B) taken into account when deciding to perform the Endopredict test?

A table showing the clinical and pathological characteristics of the patients classified as high- or low-risk by EPclin score should be included in the text.

Table 3. This is a very interesting table. Please indicate what do you mean by ‘borderline’ EPclin. In patient 4 (41 y) tumor size is not available although it is indicated that corresponds to a pT1c stage. Could you please clarify this point? In the same patient, Ki67 was not available. Considering the importance of Ki67 in such cases, please indicate the reason why this data is not available. I suggest, if possible, to perform the technique again.

There are minor grammatical errors in the text that should be reviewed and corrected, e.g., line 55, have instead of has; line 153, did instead of does.

6. PLOS authors have the option to publish the peer review history of their article (what does this mean?). If published, this will include your full peer review and any attached files.

Reviewer #1: No

Reviewer #2: No

Reviewer #3: No

---

## [Author Response · Author response to Decision Letter 0]

24 Dec 2019

Response to Reviewers

Reviewer #1

In this manuscript, the authors come up with EndoPredict assay in a prospective decision impact study of Mexican premenopausal breast cancer patients to change therapeutic management. The validity of EndoPredict test to assist the clinical decision-making process in premenopausal patients deserve deeper discussion. It would be helpful to observe the study longer for patient outcome and expand the sample size. Besides, the analysis of post- EndoPredict results should be illustrated more detailedly.

- Thank you for your observations. Median follow-up time was 22.9 months. During this period, four patients presented disease recurrence, with follow-up times to recurrence of 11.7, 14.3, 36.9, and 42.4 months. Two of these patients were classified as low-risk by the EndoPredict test and had local recurrences. Of the other two patients, classified as high-risk, one had a local recurrence and the other one had a distant recurrence. All patients were alive up to their last follow-up visit (lines 178-182).

- Regarding the suggestion of expanding the sample size, post-hoc extension of this prospective study which achieved its primary objective would be difficult to justify and to execute as it would need additional non-available budget and 1-2 extra years. Therefore, it is not feasible for us to do it. Hopefully, the scientific and medical community will find our results focused on an understudied population of premenopausal women valuable and significant.

- A new Table 2 and Figures 1, 2 and 3 have been added and will contribute to better illustrate the EndoPredict results.

o In Table 2, we described patients’ clinical and pathological characteristics according to their EPclin risk.

o In Figure 1, we graphed the percentage of change in chemotherapy recommendation.

o In Figures 2 and 3, we included box plots showing the chemotherapy and endocrine therapy regimens’ intensity according to the EPclin scores, as suggested by Reviewer #2.

Reviewer #2

Villareal-Garza and co-authors present an interesting decision impact study of EndoPredict in specifically younger women. The manuscript is well written and presents interesting findings on pre- versus post-treatment decision making in premenopausal women. As the authors state most validation and decision-impact studies have been performed in older, postmenopausal women and therefore their results are relevant for the clinical decision making in younger women.

A few minor points might improve the manuscript:

1. It would be helpful to add EPclin scores (means) or percentage low/high risk to Table 1 so the reader has all the information in one place.

- Thank you for this suggestion, it has been added to Table 1.

2. Table 2 would benefit from adding percentage change. I am aware that this is stated in the text but with added percentages the reader has the full picture just looking at the table.

- We have added the percentage of change in chemotherapy recommendation to Table 3 (which was previously Table 2).

3. Perhaps a Figure would make Table 2 even more appealing to readers showing the changes in treatment between pre- and post EndoPredict testing.

- The data of this table has been depicted in a new Figure 1.

4. Table 4 and 5 again please add percentages as number scan be deceiving.

- Thank you for your observation, we have included the corresponding percentages to Tables 5 and 6 (which were previously Tables 4 and 5).

5. For Table 5, I suggest putting the low intensity ET regimes, tamoxifen 5y and tamoxifen 2-3y then AI, underneath each other or to make it very clear in the table which are the low versus high intensity regimens.

- We have now specified which are low- and high-intensity regimens in Table 6 (which was previously Table 5).

6. Figures 1 and 2: I suggest showing box plots for the EPclin scores with 95% CI so it is clear to the reader what the means are and it is visually easier to see that there is a difference.

- Following your suggestion, Figures 2 and 3 (which were previously Figures 1 and 2) now show box plots.

Reviewer #3

Study design. Eligible participants (line 73). Specific selection criteria should be detailed in the text.

- Thank you for this observation. Selection criteria are now detailed in Study design (lines 73-77): Premenopausal patients with HR-positive, HER2-negative, T1-T2, and N0-N1 BC, who had not undergone systemic treatment, were eligible to participate in this study. Patients were classified as premenopausal if they had regular menses or serum FSH and estradiol levels in premenopausal ranges in case of amenorrhea <12 months of duration or irregular menses during the last year.

Line 83. The authors indicate that consensual therapeutic decisions before and after EndoPredict results were registered. However, the criteria followed when deciding the type of adjuvant treatment in each case (chemotherapy or endocrine therapy) are not specified. Please, indicate these criteria in the text. 

- These criteria have been included in the text (lines 88-91): The type of adjuvant treatment was decided according to an estimation of the risk of recurrence based on the clinical, pathological and molecular information of each particular case. More intensive or prolonged schemes were considered in high-risk patients, while less intensive or shorter ones were proposed in low-risk patients.

I assume that EndoPredict test was performed centrally and not in each of the hospitals participating in the study. If so, please include this information in the text. Information regarding the time between the test request and the result of the test should also be detailed. 

- EndoPredict tests were performed locally and results were available in a median of 18 calendar days (line 80).

Did the multidisciplinary tumor board make a provisional treatment decision-either to have or to omit adjuvant chemotherapy- in these cases? 

- Initial consensual therapeutic decisions were agreed on prior to EndoPredict results, and a second consensus was reached once results were available. The therapeutic recommendation was presented to the patient after this final decision (lines 85-87).

Did patients participate in decision making? 

- Patients did not participate in the decision-making process.

Cohort description (line 102). Tumor subtype (histology) should be detailed in the text/table 1. Please, also include the distribution of cases regarding estrogen receptor positivity (should be 100%!) and, more interesting as is usual more heterogeneous than estrogen receptor, progesterone receptor positivity/negativity. 

- Thank you for your suggestions. Tumor subtypes and estrogen and progesterone receptor positivity, with 1% cutoff, are now included in Tables 1 and 2.

The inclusion of a patient with a pT1a tumor is striking. Could you please give explain why this patient was included in the series (high tumor grade, lymph node mets…)? 

- This patient was included because she met inclusion criteria given that she had a hormone receptor-positive, HER2-negative, T1-T2, and N0-N1 breast cancer, and had not undergone systemic treatment. Moreover, her tumor was of intermediate grade and had lymphovascular invasion, which posed doubts regarding her prognosis and made the genomic test result valuable for treatment recommendations.

Was the molecular phenotype of the tumors (mainly, luminal A and luminal B) taken into account when deciding to perform the Endopredict test? 

- This data was not considered, only inclusion criteria were taken into account. However, the classification of Luminal A-like and Luminal B-like was considered for the pre- and post-test therapeutic decisions. 

A table showing the clinical and pathological characteristics of the patients classified as high- or low-risk by EPclin score should be included in the text.

- This information has been included in a new Table 2.

Table 3. This is a very interesting table. Please indicate what do you mean by ‘borderline’ EPclin. 

- By borderline EPclin, we refer to two results (3.29 and 3.32) that were very close to the cutoff point that divides the low- and high-risk categories (3.3). This clarification has been added to Table 4 (which was previously Table 3).

In patient 4 (41 y) tumor size is not available although it is indicated that corresponds to a pT1c stage. Could you please clarify this point? 

- Upon this commentary, we checked this patient’s medical record again and recovered her tumor size, which was 20 mm and indeed corresponded to a pT1c stage. This has been added to Table 4 (which was previously Table 3).

In the same patient, Ki67 was not available. Considering the importance of Ki67 in such cases, please indicate the reason why this data is not available. I suggest, if possible, to perform the technique again. 

- This patient came to our clinic after her surgery, the immunohistochemistry was performed elsewhere, and the Ki67 result was not executed there. The Ki67 testing cannot be performed again given that her biological sample is no longer available in the pathology repository.

There are minor grammatical errors in the text that should be reviewed and corrected, e.g., line 55, have instead of has; line 153, did instead of does. 

- Thank you for this observation. We have proofread the manuscript again and corrected these and a few other errors.

---

## [Decision Letter · Decision Letter 1]

27 Jan 2020

Change in therapeutic management after the EndoPredict assay in a prospective decision impact study of Mexican premenopausal breast cancer patients

PONE-D-19-24535R1

Dear Dr. Villareal-Garza,

We are pleased to inform you that your manuscript has been judged scientifically suitable for publication and will be formally accepted for publication once it complies with all outstanding technical requirements.

With kind regards,

Hakan Buyukhatipoglu

Academic Editor

PLOS ONE

---

## [Editor Report · Acceptance letter]

27 Feb 2020

PONE-D-19-24535R1 

Change in therapeutic management after the EndoPredict assay in a prospective decision impact study of Mexican premenopausal breast cancer patients 

Dear Dr. Villarreal-Garza:

I am pleased to inform you that your manuscript has been deemed suitable for publication in PLOS ONE. Congratulations! Your manuscript is now with our production department. 

With kind regards,

on behalf of

Dr. Hakan Buyukhatipoglu 

Academic Editor

PLOS ONE